# A Multi-Center, Multi-Vendor Study to Evaluate the Generalizability of a Radiomics Model for Classifying Prostate cancer: High Grade vs. Low Grade

**DOI:** 10.3390/diagnostics11020369

**Published:** 2021-02-22

**Authors:** Jose M. Castillo T., Martijn P. A. Starmans, Muhammad Arif, Wiro J. Niessen, Stefan Klein, Chris H. Bangma, Ivo G. Schoots, Jifke F. Veenland

**Affiliations:** 1Department of Radiology and Nuclear Medicine, Erasmus MC, 3015 GD Rotterdam, The Netherlands; m.starmans@erasmusmc.nl (M.P.A.S.); a.muhammad@erasmusmc.nl (M.A.); w.niessen@erasmusmc.nl (W.J.N.); s.klein@earasmusmc.nl (S.K.); i.schoots@erasmusmc.nl (I.G.S.); j.veenland@erasmusmc.nl (J.F.V.); 2Faculty of Applied Sciences, Delft University of Technology, Lorentzweg 1, 2628 CJ Delft, The Netherlands; 3Department of Urology, Erasmus MC, 3015 GD Rotterdam, The Netherlands; c.h.bangma@erasmusmc.nl; 4Department of Medical Informatics, Erasmus MC, 3015 GD Rotterdam, The Netherlands

**Keywords:** prostate carcinoma, radiomics, machine learning, MRI

## Abstract

Radiomics applied in MRI has shown promising results in classifying prostate cancer lesions. However, many papers describe single-center studies without external validation. The issues of using radiomics models on unseen data have not yet been sufficiently addressed. The aim of this study is to evaluate the generalizability of radiomics models for prostate cancer classification and to compare the performance of these models to the performance of radiologists. Multiparametric MRI, photographs and histology of radical prostatectomy specimens, and pathology reports of 107 patients were obtained from three healthcare centers in the Netherlands. By spatially correlating the MRI with histology, 204 lesions were identified. For each lesion, radiomics features were extracted from the MRI data. Radiomics models for discriminating high-grade (Gleason score ≥ 7) versus low-grade lesions were automatically generated using open-source machine learning software. The performance was tested both in a single-center setting through cross-validation and in a multi-center setting using the two unseen datasets as external validation. For comparison with clinical practice, a multi-center classifier was tested and compared with the Prostate Imaging Reporting and Data System version 2 (PIRADS v2) scoring performed by two expert radiologists. The three single-center models obtained a mean AUC of 0.75, which decreased to 0.54 when the model was applied to the external data, the radiologists obtained a mean AUC of 0.46. In the multi-center setting, the radiomics model obtained a mean AUC of 0.75 while the radiologists obtained a mean AUC of 0.47 on the same subset. While radiomics models have a decent performance when tested on data from the same center(s), they may show a significant drop in performance when applied to external data. On a multi-center dataset our radiomics model outperformed the radiologists, and thus, may represent a more accurate alternative for malignancy prediction.

## 1. Introduction

Prostate cancer (PCa) is the most common malignancy and second leading cause of cancer-related death in men [1]. From all patients diagnosed with PCa, those with low-grade lesions might be candidates for active surveillance, whereas patients with high-grade PCa require treatment [2]. The gold standard for PCa assessment in current clinical practice is histopathological verification of biopsy cores [2]. These cores are evaluated by a pathologist and assigned a grade using the Gleason score (GS). However, this procedure has shown to be susceptible to under-diagnosis of high-grade PCa and over-diagnosis of low grade PCa [3].

Multi-parametric magnetic resonance imaging (mpMRI) has received increasing interest for diagnosing, monitoring and treatment follow up for PCa. MpMRI allows non-invasive visualization of the whole prostatic tissue and extraction of quantitative parameters such as tissue density and permeability. To evaluate mpMRI, radiologists use the Prostate Imaging Reporting and Data System (PIRADS) v2, with a grading scale from one (highly unlikely to be clinically significant prostate cancer) to five (highly likely to be clinically significant prostate cancer) [4]. Nevertheless, mpMRI interpretation is challenging and prone to inter- and intra-reader variability among expert radiologists [3].

By extracting multiple imaging features, radiomics has the potential to evaluate the mpMRI data in a more objective way. In the context of PCa, the literature has shown evidence of the potential of radiomics in classifying PCa lesions [5,6,7,8], with promising performances in terms of sensitivity and specificity [9]. Nevertheless, current studies on prostate MRI radiomics still lack the quality required to allow their introduction in clinical practice [9,10]. This is due to the fact that most of the radiomics studies validated their approach by splitting their original dataset in training and validation subsets, while only a few studies performed a validation using an external set [11,12,13]. The latter evaluation is more relevant for a clinical context, where new data can present variations that were not taken into account when the original model was created. Three sources of variations can be identified: at the patient level, at the level of the MRI scanner, and at the level of the clinician. At the patient level: a model created with patient data collected in a specialized treatment centre, will differ from a model based on data collected in a hospital with a surveillance function. Magnetic resonance (MR) images vary between vendors and between scanner types from the same vendor, even if the same acquisition parameters are used. Current evidence shows that is possible to overcome these differences by applying feature harmonization techniques [14]. These techniques aim to estimate the statistical differences between imaging features computed from different data sets and apply a correction for it. To our knowledge there is no scientific evidence reporting the usage of feature harmonization in the context of PCa classification. At the clinician level: the pathologist reports, which are used as ground truth for the model, are based on the visual Gleason grading of pathologists, who are prone to considerable inter-observer variation [15,16]. Therefore, the question arises what performance can be expected when testing radiomics models on unseen multi-center-multivendor data: how generalizable are radiomics model in the context of PCa? The number of studies addressing generalizability is limited. To our knowledge, few studies tested their model’s generalizability for PCa detection regarding tumor aggressiveness using multiple scanners [17,18,19]. Only a few studies have validated their methods using external datasets for PCa tumor grade prediction [9]. When radiomics models are being considered as decision support tools for clinical practice, the generalizability issue should be addressed.

The main contribution of this study is two-fold. First, we assessed the generalizability of a radiomics approach for classifying PCa in a multi-center, multivendor setting. Second, in the same setting we compared the classification performance of radiologists to the performance of our radiomics model.

## 2. Materials and Methods

Our patient cohort was obtained from three healthcare centers in the Netherlands in the context of the Prostate Cancer Molecular Medicine project (PCMM), in Table 1 some of the clinical variables of this set are summarized. A Kruskal–Wallis test was performed to check whether the median of the GS distribution, volume, and prostatic specific antigen (PSA) of the included data sets were comparable.

The data usage of this study was approved by the medical ethics review committee of Erasmus MC under the number NL32105.078.10. In this PCMM-project, the mpMRI and pathology data of men with localized PCa who were scheduled for prostatectomy were prospectively collected from 2011 to 2014. In this study, we will refer to the data from the respective centers as data set A, B and C. The data of each center were visually graded by a radiologist and a pathologist working at that center. In total we included 107 patients for whom MRI, pathology images and reports were available. The distribution was as follows: A = 29, B = 38 and C = 40, the details regarding the MRI scanners and acquisition parameters of each set are described in Appendix A. The dataset shows considerable variability, with images acquired with scanners from three different vendors, using various voxel sizes and b values for the diffusion weighted sequences. In deriving our radiomics models we included the T2-weighted (T2w) and the diffusion weighted imaging (DWI) sequences and the apparent diffusion coefficient maps (ADC) derived from the DWI images.

All 107 patients had their prostate surgically removed. After the prostatectomy, the prostate was cut into 3 mm thick slices. Of the top of each slice, a photograph was taken, and 4µm coupes were cut and stained with H&E. Based on the H&E, the pathologist marked the areas with cancerous tissue on the photographs and assigned a GS to each tumor region. In Figure 1 the number of lesions per GS found in each set is summarized. We grouped lesions with a GS ≤ 6 as low-grade tumors and lesions with a GS ≥ 7 as high-grade tumors. Out of the 107 patients, 204 lesions in total were processed, 92 (45%) low-grade and 112 (55%) high-grade. The methods used to correlate the lesions found in the pathology with MRI are explained in the following section.

### 2.1. Ground Truth Construction: Pathology-MRI Correlation

A mask of identified lesions based on microscopy analysis (H&E staining) was manually drawn by a pathologist on the prostatectomy specimens’ photos. Using in house software implemented in Mevislab (v-2.2.1, Germany) [20], the macroscopy images of the prostatectomy specimen were manually registered and stacked to generate a prostate volume to enable the registration with MRI. Then, based on the prostate borders, prostate masks were manually drawn on the MR and macroscopy images. Afterwards, these two masks were manually aligned in 3D by rotation, translation, and scaling of the pathology volume. Subsequently, the translation in slice-direction was fine-tuned while inspecting the pathology and the corresponding T2w slices. As the last step, the lesion segmentation from the pathology volume was overlaid on the T2w volume.

### 2.2. Image Pre-Processing.

In order to address the variation in image resolution between and within data sets, the MR images were resampled to a voxel grid of 0.27 mm × 0.27 mm × 3 mm, which was the spacing used in the largest proportion (36%) of the T2w images.

### 2.3. Radiomics Generalizability Evaluation

To assess the generalizability of our radiomics models, we used the experimental setup as shown in Figure 2. Image data from a single center was used to train a radiomics classifier for each center. On this training set, an 100× internal random-split cross-validation was used to assess the single center performance. Finally, the model was evaluated using the other two sets to assess the generalizability; this procedure was repeated with each set. The details regarding the development of the radiomics classifiers are explained in the following section.

### 2.4. Radiomics Model Development

To generate the radiomics classifiers for each data set, we used the open-source Workflow for Optimal Radiomics Classification (v-3.3.2, Rotterdam, The Netherlands,) platform (WORC) with the default settings [21] and another setting including feature harmonization with ComBat [22]. WORC performs an automatic search amongst a wide variety of algorithms and their corresponding parameters to determine the optimal combination that maximizes the prediction performance on the training set, a schematic overview of the method is shown in Figure 3**.** The workflow starts with the user defining a region of interest (ROI) from the image, which in our case was the delineation obtained by the pathology–MRI correlation. Within these tumor masks, features quantifying intensity, shape, texture and orientation were extracted from the T2w, ADC and the highest b-value image available from the DWI images. Following feature extraction, a decision model was created, which in WORC consist of several steps, such as feature selection, oversampling and machine learning methods. WORC automatically optimizes the radiomics pipeline: during each iteration WORC generates 100,000 workflows by using different combinations of methods and parameters. At the end of each cross validation, the 50 best performing solutions were combined in an ensemble as a single classification model. The final ensemble of 50 classifiers is the resulting radiomics model, the performance of which is evaluated on the independent test set (external evaluation). Feature selection was done to select the most predictive features through enabling/disabling entire families of features (e.g., shape, local binary patterns, texture based on grey-level co-occurrence matrices). The code utilized for these experiments is available online in a GitHub repository [23].

### 2.5. Radiomics Classifier Evaluation

The internal evaluation of the model was performed by using a 100× random-split cross validation: First, the data set was split into 80% for training and 20% for testing. After this, 20% of the training set was used as validation set. This validation set was used in each training iteration to select the best parameters in order to optimize the prediction accuracy. The remaining 20% was used for performance evaluation: area under the curve (AUC), receiver operating characteristic (ROC) curve, sensitivity, and specificity. The high-grade tumors were considered the positive class. To compute the 95% confidence intervals (CI) in the cross-validation experiment, we used the corrected resampled t-test [24]. ROC confidence bands were constructed using fixed-width bands [25].

To analyze the impact of having multiple lesions from the same patient, we performed the external evaluation both at the lesion and patient level. At the patient level, for each patient only the highest grade lesion was taken into account.

### 2.6. Comparison of Our Radiomics Model with the Clinical Assessment using PIRADS v2

To compare the classification performance of a multi-center radiomics model with the clinical assessment using the PIRADS v2 score, a test set was evaluated by both radiomics and the radiologist, see Figure 4. The PIRADS scoring of the lesions was done by two radiologists with 4 years and 10 years of experience, respectively, from of the partaking centers A and B, fully blinded from histopathology results. The lesions graded as having a PIRADS ≥ 3 were considered positive for high-grade PCa and the lesions with a score ≤ 2 as negative for high-grade PCa.

For this experiment, in order to avoid a bias towards a single center, we created a test set (D) by randomly selecting 20% of the data from each of the three centers. From this set, the lesions that were not detected by one of the two radiologists were removed from the study since our goal was to compare the classification performance, not the detection rate. Subsequently, the remaining patient data (ABC*) was used to train a radiomics model to classify the patients in set D. The end performance for either radiologist and the radiomics model was computed on patient level classification.

## 3. Results

Statistical analysis of clinical variables:

The median of the Gleason Score (H = 4.63, *p* = 0.09), the lesion volume (H = 5.85, *p* = 0.06) and PSA (H = 1.99, *p* = 0.36) were similar for the three data sets.

Radiomics model generalizability:

Table 2 shows the results for the generalizability test. Overall, it can be seen that even though reasonable performances in terms of AUC (mean = 0.75) were obtained from the internal cross-validations, when the models were tested on the other data sets, the performances dropped considerably (mean AUC = 0.54). The inclusion of feature harmonization with ComBat did not improve the performance of the radiomics models. The performance metrics on the external validation sets were comparable when evaluated lesion and patient wise. Meanwhile, radiologists’ performance (mean AUC = 0.47) shows high sensitivity with a low specificity.

### Comparison of Our Radiomics Model with the Clinical Assessment using PIRADS v2

The resulting test set was composed of 16 patients with high-grade lesions and eight patients with low-grade lesions. Table 3 presents the results of the classification performance for the internal cross-validation and the performance on the test set (ABC*) for the model and the two radiologists. It can be seen that the radiomics model outperformed (AUC = 0.75) the radiologist classification with the PIRADS score (AUC of 0.50 and 0.44). Radiologists achieved a decent sensitivity (0.76 and 0.88), but near-zero specificity (0.25 and 0.0), whereas the radiomics model achieved a sensitivity of 0.88 and a specificity of 0.63.

## 4. Discussion

The expanding usage of prostate MRI for PCa diagnosis has brought an increased interest in radiomics research for tumor classification. As a result, many approaches have been proposed, and promising results have been presented, thus raising the opportunity of using these models in daily clinical workflow. However, there is limited evidence regarding the performance of these models with unseen data in a new clinical contexts, for instance with MR scanners from different vendors and/or grading by different pathologists and/or different patient profiles. Investigating how these changes affect radiomics performance is required prior to applying these models in a clinical setting.

In this study we developed radiomics classifiers starting from three independent sets and evaluated the performance on the unseen data of the other centers. To compensate for the differences between data sets and reduce the negative effects on performance that these differences might have, resampled all the images in our experiments to the same voxel size, and used the same method to correlate the pathology data to the MR data. Furthermore, we applied techniques such as normalization and class unbalance correction. While obtaining a decent performance working with data from a single center, our results showed a substantial decline in performance when evaluating the radiomics models on external data. Thus, since an internal validation on a single-center dataset is not representative of external performance, it is advisable to carry out external validations to have a realistic estimation of predictive power.

The decline in performance is most probably related to several factors. One important factor affecting the feature computation is the dependency of the radiomics features on MR scanning parameters [26]. It has been shown that image normalization applied with variety of approaches or pre-filtering cannot overcome the scan-feature dependency problem [27]. Recent literature shows evidence that it is possible to overcome the scanner-feature dependency issue by applying feature harmonization techniques such as ComBat [22]. In our experiments, we applied feature harmonization using ComBat, however the inclusion of this technique did not improve our results while testing on the external sets.

Another factor is that the delineations on the pathology data were carried out by different pathologists working at the different centers. These delineations were transferred to the MRI, but the delineation is a factor that influences the feature computation [28], compromising the likeness of the features computed from different datasets. In clinical practice, the delineation of lesions in MRI is mostly performed by a single clinician, which makes it unfeasible to test feature robustness for several delineations. Furthermore, manual delineation by specialists is time consuming and potentially subject to observer variability. Utilizing either assisted or fully automatic segmentation methods available [29,30] for the prostate and PCa lesions could improve feature computation consistency, important for radiomics approaches, and positively impact the model generalizability.

Various studies have assessed the use of radiomics in PCa classification on mpMRI [9]. To our knowledge, this is the first study to specifically address the generalizability of radiomics models in the context of PCa classification. Our study consisted of multi-centric data sets: image data from multiple vendors and multiple scanners from the same vendor, two different radiologists diagnosing the patients, three different pathology departments grading histology slices of prostatectomies as ground truth. There are studies in which one factor is varied, e.g., the study published by Dinh et al. [31]. In their study they developed a model specifically for peripheral zone PCa detection, maintaining the model’s performance between two MR scanners belonging to different vendors. However, in their experiments the data were acquired from the same center, evaluated, and processed by the same radiologists and pathologists. This might have affected positively the performance of their method.

When comparing our radiomics model to the PIRADS v2 scoring by radiologists, our results show that the radiologists achieved high sensitivity at the cost of a low specificity, while our model increased specificity substantially. This high sensitivity with PIRADS v2 may translate in clinical practice in overdiagnosis and overtreatment. A radiomics model may not only provide a more objective quantitative support tool to recommend surveillance for those cases where treatment may not instantly be required, but should also maintain a high sensitivity for those cases with aggressive PCa. However, it is important to take into account the data that the radiomics model was developed on, and the setting the model will be applied in. In other words, the safe utilization of a radiomics model in the clinic is feasible, as long as the population on which it is applied, holds similar characteristics to the population used to develop the model.

Our study has some limitations. First, our ground truth tumor grading is based on one pathologist per center, which can cause discrepancies in lesion delineations and grading. Having a consensus ground truth could have positively impacted our performance. However, this limitation represents current clinical practice, where the reader agreement between pathologists is between 70–80% [15,16].

Secondly, the number of patients included per medical center is limited. However, the total number of patients in our study is higher than the average value of 80 patients found in similar radiomics studies [9]. Thirdly, the clinical assessment was performed using the PIRADS classification v2.0 because v2.1 was not available at the moment of the readings.

Finally, we did not include clinical variables or epidemiological factors in our model. This information plays a role in clinical decision making, therefore, including this information may have a positive impact on the end performance in a multi-center and multi-vendor setting. Although, clinical patient information such as the level of PSA, the patient risk group and the outcome of the digital rectal examination were not available for a substantial number of patients which represented an obstacle to include these variables.

Despite the previous limitations, our study contributes to the field of PCa classification using radiomics by: (1) being the first study with the generalizability of PCa classification radiomics models as main focus; (2) making our scientific code available in a public repository. As regards this last point, we would like to invite the scientific community to test this code on their own data sets and so promote discussions and future collaborations.

Additionally, we would like to make some recommendations for future work: when developing a generalizable radiomics model for PCa classification the data should represent the variation present in the clinical practice with data of several centers with various pathologists and radiologists, and multiple MRI scanners from multiple vendors. The validation of the model should be performed in a prospective cohort.

## 5. Conclusions

In this paper we assessed the generalizability of radiomics models in the context of PCa grading. When limited to a specific center or, e.g., to a specific scanner or specific setting, these models perform well and may represent a valuable tool to differentiate low-grade from high grade tumors. However, when applying radiomics on data from different centers and/or scanners, a considerable drop in performance can be expected, making these models less reliable in this context.

To become clinical viable and support clinical decision making, training and validation of radiomics models should be performed in multi-center scenarios with data representative of the population on which the model will be applied.

## Figures and Tables

**Figure 1 diagnostics-11-00369-f001:**
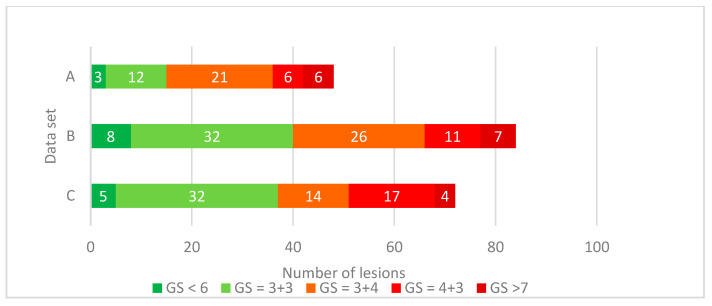
Distribution of Gleason grading of identified lesions at radical prostatectomy specimen of three different centers. The number of lesions per group is shown in white.

**Figure 2 diagnostics-11-00369-f002:**
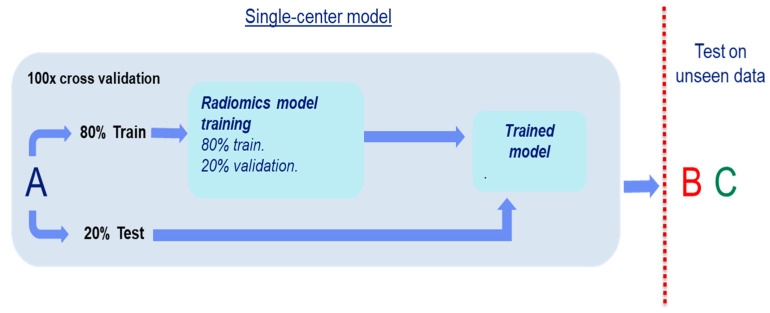
Scheme of the generalization experiment setting. In this example dataset A is used to develop a model. The model is tested on the other two sets (B and C).

**Figure 3 diagnostics-11-00369-f003:**
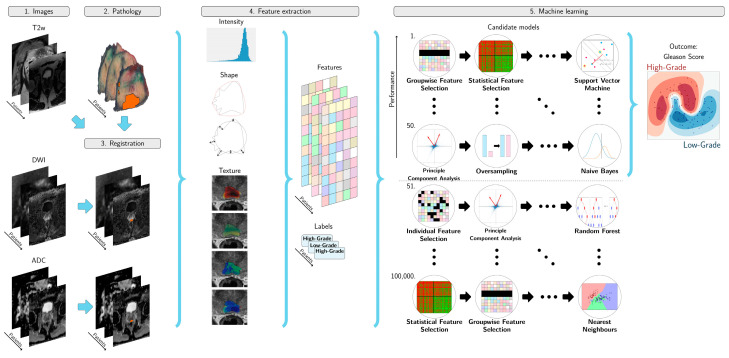
(1) The magnetic resonance sequences to be used in the model are defined. (2) The lesions from the pathology are copied and registered to the T2w sequence. (3) The diffusion weighted imaging (DWI) and apparent diffusion coefficient (ADC) are resampled and registered to the T2w. (4) Features are extracted from the T2w, DWI and ADC. (5) A radiomics model is created from the features, using an ensemble of the best 50 workflows from 100,000 candidate workflows, where the workflows are different combinations of the different classifiers.

**Figure 4 diagnostics-11-00369-f004:**
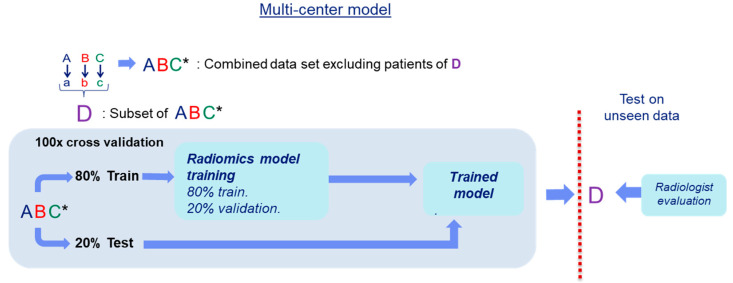
Scheme of the comparison experiment of our multi-center radiomics model with the evaluation by the radiologist. A randomly selected set of patients in ABC was set apart as test set (D), the rest of the data (ABC*) was used to develop the multi-center radiomics model.

**Table 1 diagnostics-11-00369-t001:** Prostate Cancer Molecular Medicine (PCMM) data set clinical variables and lesions characteristics. PIRADS grading performed by radiologist 1 (R1) and 2(R2). Age of patients for data sets B and C was not available (NA). PZ: Peripheral zone. TZ: transition zone. AFS: anterior fibromuscular stroma. IQR: interquartile range.

Prostate Cancer Molecular Medicine Data set Clinical Variables			
**Center**	**A**	**B**	**C**
**Number of Patients**	29	38	40
**Age at Diagnosis (mean ± std years)**	64 ± 7	NA	NA
**PSA before treatment (mean ± std ng/mL)**	12 ± 10	9 ± 5	10 ± 8
**Lesions Characteristics**			
**Number of lesions**		204	
**Lesion location**			
**PZ**	33	59	45
**TZ**	15	23	26
**AFS**	NA	2	1
**Lesion volume (median and IQR mL)**	1.6 (0.2–1.8)	1.4 (0.1–1.5)	0.8 (0.2–1.1)
**Radiologist PIRADS grading**		**R1**	**R2**
	**I**	0	4
	**II**	16	9
	**III**	21	36
	**IV**	33	34
	**V**	43	61
	**Total**	113	144

**Table 2 diagnostics-11-00369-t002:** Generalization study results. Internal: internal evaluation was performed using a 100× random-split cross-validation, reported with confidence interval. External: by training in one dataset, testing on the two remaining datasets. LC: lesion level classification. PC: patient level classification. AUC: area under the curve. CH: Test result using ComBat feature harmonization. R1 and R2: radiologist 1 and 2.

Model	Internal	External LC	External CH	External PC	R1 and R2
**Trained on A **	**A**	**B and C**			
AUC	0.75 (0.58–0.92)	0.43	0.49	0.55	0.44
Sensitivity	0.91 (0.82–1.00)	0.80	0.78	0.81	0.80
Specificity	0.30 (0.03–0.55)	0.22	0.27	0.21	0.06
**Trained on B**	**B**	**A and C**			
AUC	0.69 (0.57–0.81)	0.60	0.57	0.55	0.50
Sensitivity	0.64 (0.47–0.80)	0.43	0.74	0.86	0.88
Specificity	0.67 (0.50–0.83)	0.62	0.38	0.25	0.13
**Trained on C**	**C**	**A and B**			
AUC	0.80 (0.68–0.92)	0.60	0.62	0.65	0.44
Sensitivity	0.74 (0.66–0.86)	0.52	0.51	0.48	0.69
Specificity	0.66 (0.50–0.82)	0.63	0.69	0.63	0.19

**Table 3 diagnostics-11-00369-t003:** Performance comparison of the multi-center radiomics model with the PIRADS score performed by two radiologists. Internal: Internal cross validation results reported with confidence intervals. AUC: area under the curve. Model: results from the multi-center model for the unseen data. R1 and R2: radiologist 1 and 2, respectively.

Metrics	Internal	Model	R1	R2
**AUC**	0.72 (0.64–0.79)	0.75	0.50	0.44
**Sensitivity**	0.76 (0.66–0.89)	0.88	0.76	0.88
**Specificity**	0.55 (0.44–0.66)	0.63	0.25	0.00

## Data Availability

Please refer to suggested Data Availability Statements in section “MDPI Research Data Policies” at https://www.mdpi.com/ethics.data managing tasks related to the PCMM data set.

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
