# Peer review of "A Multi-Center, Multi-Vendor Study to Evaluate the Generalizability of a Radiomics Model for Classifying Prostate cancer: High Grade vs. Low Grade"

_diagnostics, 2021, doi:10.3390/diagnostics11020369_

Round 1

Reviewer 1 Report

The present work is focused on a rather hot topic and asks an appropriate question that remains to be solved. In light of the publication bias and the great enthusiasms around Radiomics, negative findings (e.g. poor generalizability) often go unpublished. In this light, I believe that this manuscript has something to add to the current literature. Nevertheless, I have some suggestions that the Authors might want to consider to increase the quality of the present work.

1) Introduction: to provide readers a broader view on Radiomics in prostate MRI, recent systematic reviews and meta-analysis should be mentioned (10.1016/j.ejrad.2020.109095; 10.1007/s00330-020-07027-w)

2) Materials and Methods: testing differences in age, PSA and GS distribution in the three populations should be performed to confirm that they are comparable in regard of clinical variables. Would it also be possible to have details regarding zonal distribution of the lesions? Lesion dimensions?

3) Discussion: while using pathology specimens to ensure a proper segmentation of tumor on MR images, this approach cannot be translated into clinical practice (segmentation would always be performed before treatment). Although this strategy fits the desing and purpose of the study, I believe that this issue should be better highlighted.

4) Discussion: a new version of PIRADS guidelines has been released (2.1). While I do not expect the performance of the radiologists to be undermined be the use of the older version, this issue should be probably added among the limitations of the study since version 2 is no longer state-of-art. Furthermore, the performance of PIRADS appears lower than what usually reported and this might depend on patient population characteristics (in clinical settings, PIRADS should discriminate between GS 3+3 and no PCa at all vs GS > 3+4). I hope I made my point clear.

Author Response

Response to the reviewers’ comments and recommendations

We would like to thank the reviewers for the constructive comments and suggestions. We modified the paper accordingly and as indicated in the detailed responses below.

Reviewer 1:

The present work is focused on a rather hot topic and asks an appropriate question that remains to be solved. In light of the publication bias and the great enthusiasms around Radiomics, negative findings (e.g. poor generalizability) often go unpublished. In this light, I believe that this manuscript has something to add to the current literature. Nevertheless, I have some suggestions that the Authors might want to consider to increase the quality of the present work.

Comment 1: 1) Introduction: to provide readers a broader view on Radiomics in prostate MRI, recent systematic reviews and meta-analysis should be mentioned (10.1016/j.ejrad.2020.109095; 10.1007/s00330-020-07027-w).   

Response 1:   We appreciate this comment, one of the studies have been cited in the revised version.

“Nevertheless, current studies on prostate MRI radiomics still lack the quality required to allow their introduction in clinical practice [9-10]. This is due the fact that most of the radiomics studies…” (page2 Line 60).

Comment 2: Materials and Methods: testing differences in age, PSA and GS distribution in the three populations should be performed to confirm that they are comparable in regard of clinical variables. Would it also be possible to have details regarding zonal distribution of the lesions? Lesion dimensions?

.

Response 2:  Thank you for this comment. We included the zonal distribution of the lesions and the lesion volume in TABLE 1. The statistical comparison of the data sets distribution is added to the methods section (line 96 page 3) and the results are added in the results section (page 7 line 243).

In the Methods section:

A Kruskal Wallis test was performed to check whether the median of the GS distribution, volume, and prostatic specific antigen (PSA) of the included data sets were comparable.”

In the Results section:

"Statistical analysis of clinical variables:

The median of the Gleason Score (H=4.63, p=0.09), the lesion volume (H=5.85, p=0.06) and PSA (H=1.99, p=0.36) were similar for the three data sets.

Comment 3: Discussion while using pathology specimens to ensure a proper segmentation of tumor on MR images, this approach cannot be translated into clinical practice (segmentation would always be performed before treatment). Although this strategy fits the design and purpose of the study, I believe that this issue should be better highlighted.

Response 3: We agree that a segmentation based on pathology specimen is not possible in clinical practice. However, this step is only required for model development and testing in the preclinical setting. Using the pathology specimen as ground truth gives the opportunity to relate the multiparametric MRI to the pathology specimen for the whole extent of the lesion in contrast to a design with biopsy as ground truth. When introducing such a method into clinic, the radiologist will not have the pathology information at his disposal but can use the system that has learned from cases based on the pathology information.

Comment 4: Discussion: a new version of PIRADS guidelines has been released (2.1). While I do not expect the performance of the radiologists to be undermined be the use of the older version, this issue should be probably added among the limitations of the study since version 2 is no longer state-of-art. Furthermore, the performance of PIRADS appears lower than what usually reported and this might depend on patient population characteristics (in clinical settings, PIRADS should discriminate between GS 3+3 and no PCa at all vs GS > 3+4). I hope I made my point clear.

Response 4: We thank the reviewer for addressing this potential limitation. We agree that PI-RADS classification is an evolutionary document, adopting insights, expertise and research results from literature and expert panels, over time. We also agree that PI-RADS scoring results are dependent on multiple factors, including the mentioned population characteristics. We acknowledge the reviewer’s point by addressing the following sentence in the discussion under limitations ( Page 10 Line 357).

“Thirdly, the clinical assessment was performed using the PI-RADS classification v2.0 because v2.1 was not available at the moment of the readings."

Reviewer 2 Report

--------------------------------------------------------------

Main Comments:

Comment 1

In the 'Abstract': authors say "using open-source machine learning software". In order to guarantee the repetability of any scientific study , it is mandatory to provide any information about procedures/tools used. Authors must provide any detail about used software.

Comment 2

In section 1. 'Introduction': authors state "while only a few studies performed a validation using an external set [9]." and "Only few studies validated their methods using external datasets for PCa tumor grade prediction [9]." Referenced paper [9] is a review, therefore it does not represent a specific example of paper that use external validation. Eliminate referenced work or call a specific literature paper that uses external validation.

Comment 3

In Section 2. 'Materials and Methods': this study considers a multi-center dataset of 107 patients and 204 lesions. This means that there are lesions coming from the same patient. To consider multiple lesions of the same patient as separate lesions (a practice often used to increase the available samples) is not conceptually wrong, but introduces potentially correlated data (because they belong to the same person) into the dataset (from which the predictive model must then be built) and this could affect the capabilities of the model. How authors ave addressed this aspect?

Comment 4

In Section 2. 'Materials and Methods', subsection 'Radiomics model development': authors state "WORC automatically optimizes the radiomics pipeline". It is possible to provide further details concerning the compliancy of this pipeline with the IBSI (Imaging Biomarker Standardization Initiative) guidelines, often used in radiomic studies?

Comment 5

In Section 2. 'Materials and Methods', subsection 'Radiomics model development': authors propose a predictive model that consists of the ensemble of several radiomic models, but does not provide any details about it. It is essential that the authors provide further details on the models that make up the ensemble and on the respective radiomic signatures used. 

Comment 6

In Section 4. 'Discussion': authors should talk about the potential use and impact (on predicitive model outcome) of multimodal methods for automatic prostate segmentation, such as the one proposed in [Rundo et al. Automated Prostate Gland Segmentation Based on an Unsupervised Fuzzy C-Means Clustering Technique Using Multispectral T1w and T2w MR Imaging. Information, 8(2), 49, June 2017, MDPI AG, doi:10.3390/info8020049, ISSN 2078-2489] that exploits on T1w and T2w series.

--------------------------------------------------------------

Minor Comments:

Comment 7

Figure 2 is not completely visible. The not visible part is intuitable (testing onto B and C dataset) and for this reason has not affected this review, but authors must fix it.

Comment 8

Section 2. Materials and Methods, subsection 'Radiomics model development': Please correct the sentence: "the maximizes de prediction performance".

Comment 9

Figure 3 is not completely visible, which has not affected this review because is a general radiomic scheme, but authors must fix it.

Comment 10

Fix Table A.1 formatting.

Comment 11

An overall English review is mandatory before publication.

Author Response

Response to the reviewers’ comments and recommendations

We would like to thank the reviewers for the constructive comments and suggestions. We modified the paper accordingly and as indicated in the detailed responses below.

Reviewer 2:

Comment 1: In the 'Abstract': authors say "using open-source machine learning software". In order to guarantee the repetability of any scientific study, it is mandatory to provide any information about procedures/tools used. Authors must provide any detail about used software.

Response 1: Thank you for this comment. We agree that including more information will facilitate the repeatability of our experiments. Therefore, we included more details about the tools used in our models as an appendix (appendix B and C) to this manuscript. More detailed documentation can be found online and the code used for this experiment is available online, we added the reference. (See reference on page 5, line 195).

“[20] josemanuel097/PCa_classification_generalizability. https://github.com/josemanuel097/PCa_classification_generalizability. Accessed 11 Feb 2021”

Comment 2: In section 1. 'Introduction': authors state "while only a few studies performed a validation using an external set [9]." and "Only few studies validated their methods using external datasets for PCa tumor grade prediction [9]." Referenced paper [9] is a review, therefore it does not represent a specific example of paper that use external validation. Eliminate referenced work or call a specific literature paper that uses external validation.

Response 2: We do not agree with this comment, we did not aim to cite a specific example. One of the main conclusions from the cited systematic review is the current lack of studies performing an external validation.

Comment 3: In Section 2. 'Materials and Methods': this study considers a multi-center dataset of 107 patients and 204 lesions. This means that there are lesions coming from the same patient. To consider multiple lesions of the same patient as separate lesions (a practice often used to increase the available samples) is not conceptually wrong, but introduces potentially correlated data (because they belong to the same person) into the dataset (from which the predictive model must then be built) and this could affect the capabilities of the model. How authors ave addressed this aspect?

Response: 3 Utilizing multiple lesions from the same patient may indeed potentially lead to correlated data. However, as lesions belonging to the same patient may have a different grade, this does not necessarily impact the capabilities of the model. As in the external validation, all lesions in the validation set are treated independently, this does not affect the performance of the model. In the internal cross-validation, there may be lesions from a patient in both the training and the test set. As the grade of the lesions may be different, the potential impact on the capabilities of the model will probably be negative. Hence, our internal cross-validation results may be pessimistic. As we focus on the external validation, and in the internal validation our model does not benefit from the potentially correlated data, we have not further addressed this in the manuscript.

Comment 4: In Section 2. 'Materials and Methods', subsection 'Radiomics model development': authors state "WORC automatically optimizes the radiomics pipeline". It is possible to provide further details concerning the compliancy of this pipeline with the IBSI (Imaging Biomarker Standardization Initiative) guidelines, often used in radiomic studies?

Response 4: Thank you for this remark. We added an appendix (Appendix B) in which we included information regarding the features used by WORC. As indicated there, WORC makes use of two feature toolboxes: PyRadiomics and PREDICT. All features from PyRadiomics are IBSI compliant: all features from PREDICT are not ISBI compliant, because the extracted features are not included in IBSI. As the scope of IBSI includes standardization of a specific set of features, there are not specific guidelines on the standardization of radiomics pipelines.

Comment 5: In Section 2. 'Materials and Methods', subsection 'Radiomics model development': authors propose a predictive model that consists of the ensemble of several radiomic models, but does not provide any details about it. It is essential that the authors provide further details on the models that make up the ensemble and on the respective radiomic signatures used.

Response 5: We agree with this comment, we added an appendix C: Adaptive workflow optimization for automatic decision model creation. In this appendix the general workflow for the optimization process is described. WORC defines a workflow as a sequential combination of algorithms and their respective parameters. To create a workflow, WORC includes algorithms to perform feature scaling, feature imputation, feature selection, oversampling, and machine learning. It is not feasible to give the details of all the workflows, such as the radiomics features and classifiers used, in this paper.

Comment 6: In Section 4. 'Discussion': authors should talk about the potential use and impact (on predicitive model outcome) of multimodal methods for automatic prostate segmentation, such as the one proposed in [Rundo et al. Automated Prostate Gland Segmentation Based on an Unsupervised Fuzzy C-Means Clustering Technique Using Multispectral T1w and T2w MR Imaging. Information, 8(2), 49, June 2017, MDPI AG, doi:10.3390/info8020049, ISSN 2078-2489] that exploits on T1w and T2w series.

Response 6: We apologize for not understanding this comment. Our study focusses on radiomics for PCa classification, and not on segmentation tools for the whole prostate. This is a different field of research, automatic prostate segmentation is outside the scope of our study. 

--------------------------------------------------------------

Minor Comments:

Comment 7

Figure 2 is not completely visible. The not visible part is intuitable (testing onto B and C dataset) and for this reason has not affected this review, but authors must fix it.

Comment 8

Section 2. Materials and Methods, subsection 'Radiomics model development': Please correct the sentence: "that maximizes de prediction performance".

Comment 9

Figure 3 is not completely visible, which has not affected this review because is a general radiomic scheme, but authors must fix it.

Comment 10

Fix Table A.1 formatting.

Comment 11

An overall English review is mandatory before publication.

Response to comment 7 ,9 and 10: We are sorry for the formatting errors. This problem was caused by differences between the template available on the Diagnostics website and the template that is used in the submission system. All the problems related to format were fixed in this revised version.

Response to comment 8: The correction was performed.

….combination that maximizes the prediction performance…

Response to comment 11:

An overall English review has been performed.

Round 2

Reviewer 1 Report

The Authors properly revised their manuscript and satisfactorily addressed all my concerns. I believe that the quality of their paper has further improved and I have no additional remarks. Kudos!

Author Response

Thank you very much for all your feedback.

Reviewer 2 Report

Comment 1

Comment 2 hasn't been addressed!

Comment 2

Comment 3 hasn't been addressed!

Comment 2a

Authors state "As in the external validation, all lesions in the validation set are treated independently, this does not affect the performance of the model."

Dependent lesions (correlated) are used as indipendent, without any assumption: statistical models for dependent data should be used. Moreover, where is it shown that it does not affect (positively or negatively) the model? Words are not enough, data must be reported!.

Comment 2b

Authors state "As the grade of the lesions may be different, the potential impact on the capabilities of the model will probably be negative. Hence, our internal cross-validation results may be pessimistic.".

These are the authors' assumptions. Nothing is reported in order to prove this. 

I suggest to the authors this literature paper [Gill et al. (2020). Correlating Radiomic Features of Heterogeneity on CT with Circulating Tumor DNA in Metastatic Melanoma. Cancers, 12(12), 3493.], which could be interesting to approach some of the problems present in this contribution.

Comment 3

Comment 6 hasn't been addressed!

I agree with the authors that the focus of their contribution is not segmentation, but mine was an advice to better understand to the potential reader how an approach of assisted segmentation methods, and in particular multimodal, could improve the repeatability of the results and how, consequently, the response of the model may vary. 

Author Response

----------------------------------------------------------------------------------------------------------------

Comment 2: In section 1. 'Introduction': authors state "while only a few studies performed a validation using an external set [9]." and "Only few studies validated their methods using external datasets for PCa tumor grade prediction [9]." Referenced paper [9] is a review, therefore it does not represent a specific example of paper that use external validation. Eliminate referenced work or call a specific literature paper that uses external validation.

Response: We do not agree with this comment, we did not aim to cite a specific example. One of the main conclusions from the cited systematic review is the current lack of studies performing an external validation.

Comment: Comment 2 hasn't been addressed!

Response: The reference to the specific studies has been added (Page 2 line 64).

“ ……. while only a few studies performed a validation using an external set [11–13]

“11.    Transin S, Souchon R, Gonindard-Melodelima C, et al (2019) Computer-aided diagnosis system for characterizing ISUP grade ≥ 2 prostate cancers at multiparametric MRI: A cross-vendor evaluation. Diagn Interv Imaging 100:801–811. https://doi.org/10.1016/j.diii.2019.06.012”

“12.    Penzias G, Singanamalli A, Elliott R, et al (2018) Identifying the morphologic basis for radiomic features in distinguishing different Gleason grades of prostate cancer on MRI: Preliminary findings. PLoS One 13:. https://doi.org/10.1371/journal.pone.0200730”

“13.    Dinh AH, Melodelima C, Souchon R, et al (2018) Characterization of Prostate Cancer with Gleason Score of at Least 7 by Using Quantitative Multiparametric MR Imaging: Validation of a Computer-aided Diagnosis System in Patients Referred for Prostate Biopsy. Radiology 287:525–533. https://doi.org/10.1148/radiol.2017171265”

----------------------------------------------------------------------------------------------------------------

Comment: In Section 2. 'Materials and Methods': this study considers a multi-center dataset of 107 patients and 204 lesions. This means that there are lesions coming from the same patient. To consider multiple lesions of the same patient as separate lesions (a practice often used to increase the available samples) is not conceptually wrong, but introduces potentially correlated data (because they belong to the same person) into the dataset (from which the predictive model must then be built) and this could affect the capabilities of the model. How authors ave addressed this aspect?

Response: Utilizing multiple lesions from the same patient may indeed potentially lead to correlated data. However, as lesions belonging to the same patient may have a different grade, this does not necessarily impact the capabilities of the model. As in the external validation, all lesions in the validation set are treated independently, this does not affect the performance of the model. In the internal cross-validation, there may be lesions from a patient in both the training and the test set. As the grade of the lesions may be different, the potential impact on the capabilities of the model will probably be negative. Hence, our internal cross-validation results may be pessimistic. As we focus on the external validation, and in the internal validation our model does not benefit from the potentially correlated data, we have not further addressed this in the manuscript.

Comment 2

Comment 3 hasn't been addressed!

Comment 2a

Authors state "As in the external validation, all lesions in the validation set are treated independently, this does not affect the performance of the model." Dependent lesions (correlated) are used as indipendent, without any assumption: statistical models for dependent data should be used. Moreover, where is it shown that it does not affect (positively or negatively) the model? Words are not enough, data must be reported!.

Comment 2b

Authors state "As the grade of the lesions may be different, the potential impact on the capabilities of the model will probably be negative. Hence, our internal cross-validation results may be pessimistic." These are the authors' assumptions. Nothing is reported in order to prove this. I suggest to the authors this literature paper [Gill et al. (2020). Correlating Radiomic Features of Heterogeneity on CT with Circulating Tumor DNA in Metastatic Melanoma. Cancers, 12(12), 3493.], which could be interesting to approach some of the problems present in this contribution.

Response 2a and 2b:

We addressed this comment by also computing our metrics on a patient level. A description of this procedure was added to the methods section and the results were added to Table 2 in the results section.  

Methods (Page 6 line 222 ) : “To analyze the impact of having multiple lesions from the same patient, we performed the external evaluation both on lesion and patient level. On the patient level, for each patient only the highest grade lesion was taken into account.”

Results (Page 7 line 260): “…..The performance metrics on the external validation sets were comparable when evaluated lesion and patient wise.”

----------------------------------------------------------------------------------------------------------------

Comment 6: In Section 4. 'Discussion': authors should talk about the potential use and impact (on predicitive model outcome) of multimodal methods for automatic prostate segmentation, such as the one proposed in [Rundo et al. Automated Prostate Gland Segmentation Based on an Unsupervised Fuzzy C-Means Clustering Technique Using Multispectral T1w and T2w MR Imaging. Information, 8(2), 49, June 2017, MDPI AG, doi:10.3390/info8020049, ISSN 2078-2489] that exploits on T1w and T2w series.

Response 6: We apologize for not understanding this comment. Our study focusses on radiomics for PCa classification, and not on segmentation tools for the whole prostate. This is a different field of research, automatic prostate segmentation is outside the scope of our study. 

Comment 3: Comment 6 hasn't been addressed!.

I agree with the authors that the focus of their contribution is not segmentation, but mine was an advice to better understand to the potential reader how an approach of assisted segmentation methods, and in particular multimodal, could improve the repeatability of the results and how, consequently, the response of the model may vary. 

Response 3: Thanks for clarifying this comment. In the discussion we addressed how the feature computation can be affected (and in consequence de model performance) by the lesion segmentation. Inside this section, based on your comment we introduce to the reader how having an automatic method to perform the delineation might positively impact the consistency of the model (page 9 line 330).

….Furthermore, manual delineation by specialists is time consuming and potentially subject to observer variability. Utilizing either assisted or fully automatic segmentation methods available [29,30] for the prostate and PCa lesions could improve feature computation consistency, important for radiomics approaches,  and positively impact the model generalizability.”  

“29.     Rundo L, Militello C, Russo G, et al (2017) Automated Prostate Gland Segmentation Based on an Unsupervised Fuzzy C-Means Clustering Technique Using Multispectral T1w and T2w MR Imaging. Information 8:. https://doi.org/10.3390/info8020049

“30.     Arif M, Schoots, Ivo G., Castillo T. Jose M., Bangma CH, et al (2020) Clinically significant prostate cancer detection and segmentation in low-risk patients using a convolutional neural network on multi-parametric MRI. Eur Radiol 1–11. https://doi.org/10.1007/s00330-020-07008-z”

-----------------------------------------------------------------------------------------------------------------

Round 3

Reviewer 2 Report

All raised comments were properly approached by authors.